## 1 Streamflow elasticity as a function of aridity

Vazken Andréassian<sup>\*,1</sup>, Guilherme M. Guimarães<sup>1</sup>, Julien Lerat<sup>2</sup>, Alban de Lavenne<sup>1</sup>

- <sup>1</sup> Université Paris-Saclay, INRAE, HYCAR Research Unit, Antony, France
- 6 <sup>2</sup> CSIRO, Canberra, Australia
- 7 \*Corresponding author, vazken.andreassian@inrae.fr

9 Abstract

4

8

22

26

- Relating variations in annual streamflow to a climate anomaly, commonly referred to
- as streamflow elasticity to climate, is central for a rapid assessment of the impact of
- climate change on water resources. This elasticity is classically estimated via a multiple
- linear regression between anomalies in streamflow and climate variables. However,
- this approach does not explicitly account for the fact that elasticity depends on aridity
- as suggested by "Budyko-type" water balance formulas. Using a large dataset of 4,122
- catchments from four continents, we first verify empirically the link between elasticity
- and aridity. Then, we propose a method to constrain elasticity coefficients with
- derivatives from a "Budyko-type" water balance formula, that allows introducing an
- explicit dependency between elasticity and aridity. We show that adding this
- dependency produces a regionalized elasticity formula with physically-realistic
- elasticity coefficients.
- Keywords: elasticity, sensitivity, aridity index, humidity index, Schreiber formula,
- Oldekop formula, Turc-Mezentsev formula, Bagrov formula, Tixeront-Fu formula,
- Budyko hypothesis, hierarchical linear model

### **Notations**

- This study uses three hydrological fluxes: precipitation  $(P_n)$ , streamflow  $(Q_n)$ , and
- potential evaporation  $(E_{0n})$ . All fluxes are computed at the catchment scale as annual
- sums, expressed in millimeters per year. The subscript n refers to a specific
- hydrological year. For the Northern Hemisphere, the hydrological year spans from 1st

- October of year n-1 to 30<sup>th</sup> September of year n. For the Southern Hemisphere, it
- spans from 1<sup>st</sup> April of year n to 31<sup>st</sup> March of year n+1. Thus,  $Q_n$ , represents the
- streamflow for the hydrological year n. Long-term mean values are denoted by an
- overbar (e.g.,  $\bar{Q}$ ). Annual anomalies, denoted by  $\Delta$ , are computed as the difference
- between the annual value and the long-term mean. For example, the streamflow
- anomaly is calculated as  $\Delta Q_n = Q_n \bar{Q}$ . This is also applied to precipitation ( $\Delta P_n = P_n \bar{Q}$ ).
- $\bar{P}$ ) and potential evaporation ( $\Delta E_{0n} = E_{0n} \overline{E_0}$ ).
- Additionally, we also define a combined flux,  $\Lambda_n$  (see Eq. 2), which reflects the
- synchronicity of precipitation and potential evaporation. This is also expressed in
- millimeters per year, and its anomalies are computed as  $\Delta \Lambda_n = \Lambda_n \bar{\Lambda}$ .
- The aridity index,  $\varphi$ , corresponds to the ratio  $\overline{E_0}/\overline{P}$ , while the inverse ratio  $\overline{P}/\overline{E_0}$
- corresponds to the *humidity index*.

### 43 1 Introduction

### 44 1.1 About streamflow elasticity

- The climate elasticity of streamflow (Schaake and Liu, 1989; Dooge et al., 1999;
- Sankarasubramanian et al., 2001) describes the sensitivity of streamflow to changes
- in a climate variable. Elasticity is classically derived from the following regression:

$$\Delta Q_n = e_{Q/P} \Delta P_n + e_{Q/E_0} \Delta E_{0n}$$
 Eq. 1

- where:  $e_{Q/P}$  denotes the precipitation elasticity of streamflow,  $e_{Q/E_0}$  denotes the
- potential evaporation elasticity of streamflow; both coefficients are dimensionless. Note
- that elasticity is defined here in absolute terms, i.e. as the sensitivity between quantities
- of the same dimension ( $\Delta Q$ ,  $\Delta P$  and  $\Delta E_0$  are all in mm/y) following Andréassian et al.
- (2016).
- Andréassian et al. (2025) recently proposed to enrich the traditional computation given
- in Eq. 1, to account for the seasonal time shift between precipitation and potential
- evaporation, because of its decisive impact on catchment water yield (see e.g. Pardé,
- 1933; Coutagne and de Martonne, 1934; Thornthwaite, 1948; Milly, 1994; Yokoo et al.,
- 2008; Roderick and Farquhar, 2011; de Lavenne & Andréassian, 2018; Feng et al.,
- 2019). We compute the synchronous amount of precipitation and potential evaporation
- $\Lambda$ , using monthly data as in Eq. 2:

$${\Lambda _n} = \frac{{\sum_{m = 1}^{12} {min\left( {{P_{m,n}} \;,{E_0}_{m,n}} \right)} }}{{\sqrt {{P_n} *{E_0}_n} }} * \bar P$$
 Eq. 2

Where index m stands for the month. The dimension of  $\Lambda_n$  is mm/y and it represents 61 the annual precipitation volume most easily accessible to evaporation. For two years 62 with the same annual amounts of precipitation and potential evaporation,  $\Lambda$  will be

higher when they are synchronous, and lower when they are out of phase (for more

details, please refer to Andréassian et al., 2025). With this new term, the regression in

Eq. 1 becomes:

$$\Delta Q_n = e_{Q/P} \Delta P_n + e_{Q/E_0} \Delta E_{0n} + e_{Q/\Lambda} \Delta \Lambda_n \tag{Eq. 3}$$

The link between streamflow elasticity and catchment aridity is a well-established

### 1.2 Using aridity to estimate streamflow elasticity

concept in hydrology, an idea that can be traced back to Oldekop (1911) and his followers, including Budyko (1948), Bagrov (1953) and Mezentsev (1955). Many 'modern' hydrologists such as Dooge (1992) and Dooge et al. (1999) discussed the form that aridity-dependent streamflow formulas could take. This dependency was emphasized by Koster and Suarez (1999), who write that "the partitioning of a precipitation anomaly into evaporation and runoff anomalies is a simple function of the dryness index", while Arora (2002) concludes that "the use of aridity index provides a straight-forward method to obtain a first order estimate of the effect of climate change on annual runoff". Chiew (2006) shows the dependency of streamflow elasticity on aridity, Renner et al. (2012) stress that the elasticity of streamflow "is largely dependent on [...] the aridity of the climate" and Roderick and Farquhar (2011) underline that "the response of runoff to changes in the main driving variables is not constant but depends on the overall climatic dryness". More recently, the concept has been applied at a global scale, with Berghuijs et al. (2017) who use the elasticity pattern provided by the Tixeront-Fu formula to propose a world map of aridity-dependent streamflow elasticities, Zhang et al. (2022) discuss the impact of aridity on the sensitivity of the elasticity coefficient to the aggregation time step, and Anderson et al. (2024) extends the computation of elasticity to different flow quantiles, and show that aridity impacts the shape of the curve relating the different elasticity quantiles.

### 1.3 Local vs class estimation of elasticity

To estimate the climate elasticity of streamflow at regional or national scales, making 90 the dependency of streamflow elasticity on aridity explicit can constrain the estimation 91 of elasticity coefficients and increase their physical realism. 92 For a given catchment with a sufficiently long series of annual observations, streamflow 93 elasticity can be computed locally by linear regression (Andréassian et al., 2016). 94 However, for ungaged catchments, local estimation of elasticity coefficients is no 95 longer possible. Instead, a class-elasticity can be estimated by combining all available 96 records in a region. The estimation by class has both advantages and drawbacks.

While this approach improves the statistical significance of elasticity coefficients, which

can have high uncertainty when estimated locally (especially for potential evaporation), it also requires combining data from catchments with different aridity indices. This

presents a challenge, precisely because we know that aridity and elasticity are linked.

Methods to estimate local- and class-elasticity are detailed in section 2.

### 1.4 Formulas relating streamflow elasticity to aridity

We mentioned above the seminal work of the hydrologists who, following Oldekop (1911), developed various mathematical formulas to represent catchment water balance. These studies established simple water balance formulas from which a "theoretical" elasticity of streamflow can be derived as their partial derivatives. In Table 1, we present four long-term water balance formulas that can be used to provide these theoretical elasticity estimates. The Schreiber and Oldekop formulas are parameter-free, while the Turc-Mezentsev and Tixeront-Fu formulas each have one parameter ( $\omega^1$  and m, respectively). These last two formulas are equivalent when setting  $m=\omega+0.72$  (Yang et al., 2008; Andréassian and Sari, 2018), which explains why their curves overlap in some of the later figures.

Table 1 also presents the partial derivatives for each formula, allowing to compute the precipitation and the potential evaporation elasticities of streamflow. Unsurprisingly,

these formulas are all functions of the aridity index.

111112

 $<sup>^{1}</sup>$  We use  $\omega$  instead of the more commonly used "n" on purpose, to avoid confusion with the subscript n used for years.

- Table 1. Common long-term water balance formulas and the associated 118 elasticities ( $\overline{Q}$  long-term average streamflow [mm/y],  $\overline{P}$  long-term average precipitation [mm/y],  $\overline{E_0}$  long-term average reference evaporation [mm/y],  $\varphi$  = 120  $\overline{E_0}$  is the avidity index)
- $\frac{\overline{E_0}}{\overline{p}}$  is the aridity index)

| Name                                               | Formula                                                                                              |        | Precipitation elasticity $\frac{\partial Q}{\partial P}$                |        | Potential evaporation elasticity $\frac{\partial Q}{\partial E_0}$                                                                                    |        |
|----------------------------------------------------|------------------------------------------------------------------------------------------------------|--------|-------------------------------------------------------------------------|--------|-------------------------------------------------------------------------------------------------------------------------------------------------------|--------|
| Schreiber<br>(Oldekop, 1911)                       | $ar{Q} = ar{P} \cdot exp\left(-rac{\overline{E_0}}{\overline{P}}\right)$                            | Eq. 4  | $e_{Q/p}={(1+\phi)}e^{-\phi}$                                           | Eq. 5  | $e_{Q}/_{E_{0}}=-e^{-arphi}$                                                                                                                          | Eq. 6  |
| Oldekop, 1911)                                     | $ar{Q}=ar{P}-\overline{E_0}$ . $tanh\left(rac{ar{P}}{E_0} ight)$                                    | Eq. 7  | $e_{Q^{/P}}=tanh^{2}\left(rac{1}{arphi} ight)$                         | Eq. 8  | $egin{aligned} &e_{Q^{\prime}E_{0}} \ &= -tanh\left(rac{1}{\phi} ight) \ &+rac{1}{\phi}\Big[1-tanh^{2}\left(rac{1}{\phi} ight)\Big] \end{aligned}$ | Eq. 9  |
| Turc-Mezentsev<br>(Turc, 1954; Mezentsev,<br>1955) | $ar{Q}=ar{P}-ar{P}^{-\omega}+\overline{E_0}^{-\omega}ig]^{-1}_{\overline{\omega}}$ with $\omega>0$   | Eq. 10 | $e_{Q^{\prime}P}$ $= 1 - (1 + \varphi^{-\omega})^{-\frac{1}{\omega}-1}$ | Eq. 11 | $e_{Q/E_0} = -(1 + \varphi^{\omega})^{-rac{1}{\omega}-1}$                                                                                            | Eq. 12 |
| <b>Tixeront-Fu</b><br>(Tixeront, 1964; Fu, 1981)   | $ar{Q} = igl[ar{P}^{ m m} + \overline{E_0}^{ m m}igr]^{rac{1}{ m m}} - \overline{E_0}$ with $m > 1$ | Eq. 13 | $e_{Q/P} = (1 + \varphi^m)^{\frac{1}{m}-1}$                             | Eq. 14 | $e_{Q}/E_{0}$ $= -1 + (1 + \varphi^{-m})^{\frac{1}{m}-1}$                                                                                             | Eq. 15 |

https://doi.org/10.5194/egusphere-2025-4912 Preprint. Discussion started: 5 November 2025 © Author(s) 2025. CC BY 4.0 License.

Figure 1 illustrates the similarities and differences among the formulas by showing their respective elasticity-aridity relationships. The embedded dependency on aridity is clearly visible, and we notice that the four formulas have distinct but similar shapes (with the difference between the Turc-Mezentsev and the Tixeront-Fu being negligible). Furthermore, the precipitation elasticity is bounded between 0 and 1, which means that one millimeter of additional precipitation will always result in less than one millimeter of additional streamflow. Similarly, the potential evaporation elasticity is bounded between 0 and -1, which means that one millimeter of additional potential evaporation will always result in a decrease of streamflow of less than one millimeter. These bounds represent a physically-realistic catchment response.

Figure 1: Theoretical relationships between streamflow elasticities and the humidity index (left panel) and the aridity index (right panel). Blue lines represent the precipitation elasticity of streamflow, and orange lines represent the potential evaporation elasticity of streamflow.

### 1.5 Purpose of this paper

This paper aims to verify empirically the fact that streamflow elasticity depends on aridity, and to show how the theoretical pattern provided by the "Budyko-type" water balance formulas can help constrain the estimation of elasticity coefficients, yielding physically-coherent regionalized streamflow elasticities. We use for this purpose a large dataset of catchments covering a wide variety of climates.

## 2 Catchments and Method

### 2.1 Test catchments

To ensure that our analysis was based the widest possible range of climates, we used 146 a set of 4,122 catchments, representing 162,005 station-years of data (average length of catchment time series is 39 years). It includes catchments from Australia (Fowler et 147 148 al., 2024), Brazil (Almagro et al., 2021), Denmark (Liu et al., 2024), France (Delaigue et al., 2024), Germany (Loritz et al., 2024), Sweden (de Lavenne et al., 2022), 149 150 Switzerland (Höge et al., 2023), the United Kingdom (Coxon et al., 2020) and the USA 151 (Addor et al., 2017). Because this dataset is exactly the same as the one used by 152 Andréassian et al. (2025), we refer the reader to this paper for the details of the 153 selection of the catchments from the original datasets. 154 In our dataset, the aridity indices range from 0.1 to 6.3, with a first quartile of 0.6 and 155 a third quartile of 1.0. The mean and the median of the aridity index are both 0.8. To 156 assess the generality of the results, we will discuss them at the global scale and also 157 by aridity classes (as defined in Table 2).

159

Table 2. Aridity classes used in this study (we only kept the classes counting more than 100 catchments)

| Aridity class | Average aridity of the class | Number of catchments | Name         |
|---------------|------------------------------|----------------------|--------------|
| [0.25,0.50[   | 0.39                         | 484                  | Very humid   |
| [0.50,0.75[   | 0.64                         | 1461                 | Humid        |
| [0.75,1.00[   | 0.85                         | 1238                 | Fairly humid |
| [1.00,1.25[   | 1.09                         | 434                  | Fairly arid  |
| [1.25,1.50[   | 1.37                         | 186                  | Arid         |
| [1.50,1.75[   | 1.61                         | 109                  | Very arid    |

168

174

179 180

181

Figure 2. location of the catchments studied and repartition by aridity classes

### 165 2.2 Computation of local elasticities

Our reference method will consist in the local (i.e., catchment-specific) computation of 167 streamflow elasticities using Eq. 16:

$$\Delta Q_n = e_{O/P}^{loc} \Delta P_n + e_{O/E_0}^{loc} \Delta E_{0n} + e_{O/A}^{loc} \Delta \Lambda_n$$
 Eq. 16

Because the elasticity coefficients are obtained through linear regression, they are associated with statistical uncertainty, which we assess using the p-value. A 169 170 significance threshold must be chosen, above which a coefficient is not considered 171 statistically different from zero. For this paper, we use a conventional threshold of 0.05. 172 With this local approach, a unique triplet of elasticities is computed for each of the 173 4,122 catchments, and the goodness of fit for each regression is, by definition, maximized (hence our choice of the local calibration as reference).

#### 175 Computation of unique elasticities by aridity class and for the entire dataset

We can also estimate a single triplet of elasticities for each different aridity class (as 177 defined in Table 2) and we then use Eq. 17 for all the catchments of the given aridity 178

$$\Delta Q_n = e_{Q/P}^{cl} \Delta P_n + e_{Q/E_0}^{cl} \Delta E_{0n} + e_{Q/A}^{cl} \Delta \Lambda_n$$
 Eq. 17

Estimating a single triplet of elasticities for each class allows investigating the dependency of elasticity to aridity. To calibrate the three parameters, we use a simple grid search algorithm. The objective function to be maximized is the bounded Nash-

182 Sutcliffe Efficiency of Mathevet et al. (2006), which is first calculated for each

183 catchment separately, and then averaged over the catchments belonging to the class

and used as the objective to maximize (see Section 2.5).

For reference, we also compute a single triplet of elasticities at the global scale by

pooling all 4,122 catchments together. By construction, this world-wide triplet yields

the lowest mean efficiency.

### 2.4 Computation of regionalized elasticities

In the regionalized approach, we use the entire dataset to calibrate a single underlying

model, similarly to the calculation of elasticities at the global scale. However, this

method ultimately produces catchment-specific results. Each catchment has a distinct

triplet of elasticities because the elasticities for precipitation  $(e_{Q/P}^{reg})$  and potential

evaporation  $(e_{0/E_0}^{reg})$  are modeled as functions of each catchment's aridity index  $(\varphi)$ ,

given by Eq. 18. The regionalization formulas are adjusted by a shape parameter noted

$\alpha$ :

188

$$\begin{split} \Delta Q_n &= e_{Q/P}^{reg} \Delta P_n + e_{Q/E_0}^{reg} \Delta E_{0_n} + e_{Q/A}^{reg} \Delta \Lambda_n \\ &e_{Q/P}^{reg} = f_P(\alpha_P, \varphi) \\ &e_{Q/E_0}^{reg} = f_{E_0} \big(\alpha_{E_0}, \varphi\big) \\ &e_{Q/A}^{reg} = constant \text{ (does not depend on } \varphi) \end{split}$$

There were several alternatives available for choosing the shape of functions  $f_P$ , and

$f_{E_0}$ , as well as for adjusting the shape parameters. For  $f_P$  and  $f_{E_0}$  we used the

derivatives of the Oldekop formula (see Eq. 8 and Eq. 9). The variation range for these

201 functions was constrained based on the results of the class calibration (Section 2.3).

The synchronicity elasticity  $(e_{O/\Lambda}^{reg})$  was kept constant because no clear empirical

relationship was observed when examining either the local or the class-calibrated

elasticities.

Figure 3 illustrate the dependency of streamflow elasticities to aridity, which is apparent

both with the locally- and the class-estimated values. To keep the number of adjusted

parameters low, we adjusted only three parameters ( $\alpha_P$ ,  $\alpha_{E_0}$  and  $e_{Q/\Lambda}^{reg}$ ) for Eq. 18, the

- variation bounds were set up empirically once for all based on the results of the class
- calibration.

### 210 2.5 Model evaluation criterion

- To evaluate the performance of the different elasticity models in simulating streamflow
- anomalies, we use the classical Nash and Sutcliffe (1970) efficiency criterion (NSE).
- The NSE is usually computed for each of the 4,122 catchments separately using Eq.
- 19:

$$NSE = 1 - rac{\sum_n (\Delta Q_n^{obs} - \Delta Q_n^{cal})^2}{\sum_n \left(\Delta Q_n^{obs} - \overline{\Delta Q^{obs}}
ight)^2}$$
 Eq. 19

- Because the NSE varies in the interval  $]-\infty,1]$ , it is not recommended to compute an
- average over large sets (indeed, a few very low criteria values will impact the average
- criterion value). For this reason, we follow Mathevet et al. (2006) and use the bounded
- form (called "C2M" in the original paper) as in Eq. 20:

Bounded NSE (C2M) = 
$$\frac{NSE}{2 - NSE}$$
 Eq. 20

### 220 3 Results

## 221 3.1 Empirical verification of the dependency between locally-estimated

### 222 streamflow elasticities and aridity

- We first computed the local streamflow elasticities for each catchment by linear
- regression (Eq. 16), and retained only the coefficients that were statistically significant
- at the 0.05 level. In our dataset, 97% of the catchments had a significant  $e_{O/P}$
- parameter, only 23% of the catchments had a significant  $e_{O/E_0}$  parameter, and 64% of
- the catchments had a significant  $e_{Q/\Lambda}$  parameter.
- Figure 3 presents the link between aridity and the locally-estimated elasticity
- coefficients. The results confirm the expected dependency between precipitation
- elasticity and aridity, which was previously shown for the theoretical formulas in Figure
- 1. For the potential evaporation elasticity, a satisfying trend is visible but many
- physically unrealistic elasticities show that additional constraints are required for this
- term. Finally, the  $\Lambda$ -elasticity of streamflow (i.e. the streamflow elasticity towards the

synchronous amounts of precipitation and streamflow), shows no clear dependency on the aridity index (but we did not expect any relationship).

Figure 3. Relationship between the aridity index and locally-estimated climatic elasticities of streamflow, for precipitation elasticity (left), potential evaporation elasticity (middle), synchronicity elasticity (right). The white domain indicates the physically-plausible range (i.e. [0,1] for precipitation elasticity and [-1,0] for potential evaporation and synchronicity elasticities. a - (upper panel) locally calibrated elasticity coefficients, all plots include only catchments with statistically significant elasticity coefficients (p < 0.05), resulting in different sample sizes for each panel (N = 4017 for  $e_{Q/P}$ , N = 957 for  $e_{Q/E_0}$  and N = 2630 for  $e_{Q/A}$ ); b – (lower panel) class calibrated elasticity coefficients (from Table 3)

5

aridity index E<sub>0</sub>/F

aridity index E<sub>0</sub>/P

### 3.2 Results by aridity class

2 3 4 5

aridity index E<sub>0</sub>/P

We also calibrated the three elasticity coefficients to obtain a single triplet of values for each of the aridity classes as defined in sections 2.4 and 3.2. The resulting class-calibrated values are presented in Table 3. As reference, the performance of the local

(catchment-specific) estimation is also provided (by construction, it represents the upper limit of performance).

Table 3. Class-calibrated elasticity values for catchments grouped by the aridity index  $\phi$ 

| Aridity class                               | Number of catchments | Elasticity values |                  |                         | Performance expressed in mean bounded NSE for                                    |                                                                             |  |
|---------------------------------------------|----------------------|-------------------|------------------|-------------------------|----------------------------------------------------------------------------------|-----------------------------------------------------------------------------|--|
|                                             |                      | $e^{cl}_{Q/P}$    | $e^{cl}_{Q/E_0}$ | $e^{cl}_{Q/\varLambda}$ | Class approach<br>(same elasticities<br>for all catchments<br>in the same class) | Reference<br>approach (local<br>i.e., catchment-<br>specific<br>estimation) |  |
| <b>Very Humid</b> $\varphi \in [0.25, 0.5[$ | 484                  | 0.72              | -0.44            | -0.36                   | 0.59                                                                             | 0.68                                                                        |  |
| <b>Humid</b> $\varphi \in [0.5, 0.75[$      | 1461                 | 0.56              | -0.22            | -0.50                   | 0.46                                                                             | 0.57                                                                        |  |
| Fairly Humid $\varphi \in [0.75,1[$         | 1238                 | 0.49              | -0.25            | -0.53                   | 0.42                                                                             | 0.52                                                                        |  |
| Fairly Arid $\varphi \in [1,1.25[$          | 434                  | 0.33              | -0.32            | -0.44                   | 0.32                                                                             | 0.49                                                                        |  |
| Arid $\varphi \in [1.25, 1.5[$              | 186                  | 0.18              | -0.19            | -0.18                   | 0.27                                                                             | 0.56                                                                        |  |
| <b>Very Arid</b> $\varphi \in [1.5,1.75[$   | 109                  | 0.17              | -0.08            | -0.20                   | 0.29                                                                             | 0.55                                                                        |  |
| World                                       | 4,122                | 0.46              | -0.19            | -0.56                   | 0.38                                                                             | 0.56                                                                        |  |

The numeric values in Table 3 confirm the tendency identified in Figure 3: the precipitation elasticity of streamflow shows a clear decreasing trend with increasing aridity, while the potential evaporation elasticity shows a symmetric increasing trend. The empirical range of variation observed in the class-calibrated results is narrower than the theoretical range from the water balance formulas: for  $e_{q/P}$ , the observed range is [0.17, 0.72] compared to the theoretical [0, 1], and for  $e_{q/P}$ , the range is [-0.44, -0.08] compared to the theoretical [-1, 0]. Finally, there is no clear trend identifiable for  $e_{q/A}$ . A clear advantage of the class-based calibration approach is that all resulting elasticities values fall, without exception, within the physically-realistic ranges.

# 3.3 Constraining the elasticity estimation with an aridity-dependent formulation: test for the entire dataset

The observed link between the aridity index and the local elasticity estimates suggested us to test the solution presented in section 2.4, using a "regionalized"

estimation of the elasticities of streamflow. This approach makes use of the identified pattern to enforce physical coherence across the entire dataset. To parameterize this relationship, we adapted the partial derivative of the parameter-free Oldekop formula (Table 1). We constrained the output of the Oldekop formulas to the empirical range observed in the class-based calibration (Table 3), offsetting the range for  $e_P$  to [0.15, 0.75], and for  $e_{E_0}$  to [-0.45, -0.10]. Thus, the regionalized elasticities are calculated as:

$$e_{Q/P}^{reg} = 0.15 + |0.75 - 0.15| * f_{P-Oldekop}(\alpha_P, \varphi)$$
 Eq. 21

where  $f_{P-Oldekop}$  is given by Eq. 8

$$e_{O/E_0}^{reg} = -0.10 + |-0.45 + 0.10| * f_{E_0-Oldekop}(\alpha_{E_0}, \varphi)$$
 Eq. 22

where  $f_{E_0-Oldekop}$  is given by Eq. 9

0.56

Note that the restricted ranges remain within the physically-realistic limits.

We can now compare the performance of three modeling approaches: the "upper reference" where elasticities are calibrated locally at the catchment scale, the regionalized approach, and a "lower reference" with elasticities calibrated at global scale. While the upper reference requires the estimation of 12,366 parameters (3 elasticities for 4,122 catchments), the latter two require only 3 parameters each. The corresponding results are presented below in Table 4.

Table 4. Results of the application of the regionalized approach to all the catchments of our dataset (4,122): the performance is compared to a upper reference (with locally calibrated elasticity values) and a lower reference (with a unique value calibrated for all the catchments in the world)

| Performance expressed in mean bounded NSE for |                          |                       |  |  |
|-----------------------------------------------|--------------------------|-----------------------|--|--|
| Upper reference                               | Regionalized approach:   | Lower reference       |  |  |
| approach:                                     | elasticities function of | approach:             |  |  |
| local, i.e. catchment-                        | each catchment's aridity | same elasticities for |  |  |
| based estimation                              | index                    | all catchments        |  |  |

There is a clear advantage for taking into account the aridity in the regionalized formula. This approach covers 28% of the performance gap between the lower and upper references, while using only three parameters. In addition, all elasticity parameters remain within the physically-realistic range. The proposed parametrization

0.43

0.38

is therefore successful from both explanatory and predictive point of views, which is a clear advantage (Andréassian, 2023).

### 4 Discussion

- In this paper, our aim was two-fold: (i) to empirically verify that at the catchment scale, streamflow elasticity and climate aridity are linked, and (ii) to propose an aridity-
- dependent parameterization allowing for the quantification of elasticity.

### 4.1 The need for an empirical verification

Because of the present popularity of Budyko's framework and its associated theoretical formulas (Table 1), an empirical verification of the elasticity-aridity link might appear superfluous. However, applying these theoretical formulas, such as the Oldekop derivative, relies on a "space-for-time-trade" assumption. This consideration assumes that a model validated across different spatial locations will also be valid for those locations for different time periods (see Peel and Blöschl, 2011, and Singh et al., 2011). Berghuijs and Woods (2016) have warned that this trade requires validation, and Berghuijs et al. (2020) stress that although the Budyko-type curves have been used to predict the evolution of catchments in response to climatic changes, they originate from "observations of spatial differences in long-term water balances, and not from observations or theory of how individual catchments respond to aridity changes". Thus, we argue that the elasticity-aridity link cannot be taken for granted and requires empirical verification, especially given the mixed results report by Oudin and Lalonde (2023).

# 4.2 An aridity-dependent parameterization that uses the shape of the Oldekop formula

Regarding our parameterization, our results confirm the general shape of the elasticity-aridity relationship given by the Oldekop formula, but they use a narrower range of variation than the theoretical one. Our work is therefore only partially coherent with the theoretical Budyko-type formulas, which appear to provide a wider range of elasticity values than our empirical data support. This should not be a surprise to hydrologists who know in particular how precipitation intensities impact the hydrological response

of arid catchments. What is remarkable, however, is that the intuition of Schreiber (1904) and Oldekop (1911), embedded in formulas of elegant simplicity, remain so useful in the 21<sup>st</sup> century. We agree on this point with Zhang and Brutsaert (2021) who suggested that the "Budyko hypothesis" could justifiably have been named after Schreiber and Oldekop, who, with so little data and only slide rules, were able to imagine tools still in use today.

### 5 Conclusion

### 5.1 Summary

In this paper, we investigated the dependency between streamflow elasticity and aridity using a large dataset of 4,122 catchments across Europe, Australia, North America and South America. Our analysis confirmed the well-established dependency between elasticity and aridity and showed that the shape of this dependency can be effectively reproduced by existing theoretical formulas. We further demonstrated that these theoretical formulas can be used to guide the regionalization process, producing a regionalized aridity-dependent estimate of streamflow elasticity for each catchment, based on a parsimonious parameterization. The proposed solution, based on the Oldekop formula, is summarized in Table 5 below and illustrated in Figure 4.

Table 5: Summary of the proposed aridity-accounting regionalized formulas for computing the precipitation and potential evaporation elasticities of streamflow.  $\Delta Q$ ,  $\Delta P$ ,  $\Delta E_0$  and  $\Delta \Lambda$  are the annual streamflow, precipitation, potential evaporation and synchronicity anomalies, respectively [mm year\*]. The nondimensional aridity index ( $\varphi = \overline{E_0}/\overline{P}$ ) is computed as a long-term average.  $f_P(\varphi)$  and  $f_{E_0}(\varphi)$  are borrowed from the Oldekop formula (see Table 1)

$$\begin{split} & \Delta Q = e_{Q/P} \Delta P + e_{Q/E_0} \Delta E_0 + e_{Q/\Lambda} \Delta \Lambda \\ & e_{Q/P} = 0.15 + 0.6 * f_P(\varphi) \\ & f_P(\varphi) = tanh^2 \left(\frac{1}{1.32 * \varphi}\right) \\ & e_{Q/E_0} = -0.10 + 0.35 * f_{E_0}(\varphi) \\ & f_{E_0}(\varphi) = -tanh \left(\frac{1}{1.43 * \varphi}\right) + \frac{1}{1.43 * \varphi} \left[1 - tanh^2 \left(\frac{1}{1.43 * \varphi}\right)\right] \\ & e_{Q/\Lambda} = -0.47 \end{split}$$

Figure 4. Regionalized relationships (from the equations in Table 5) for the climatic elasticities of streamflow as a function of the aridity index: precipitation elasticity (left), potential evaporation elasticity (middle), synchronicity elasticity (right). The white domain indicates the physically-plausible range, i.e. [0,1] for precipitation elasticity and [-1,0] for potential evaporation and synchronicity elasticities.

### 5.2 Limitations and perspectives

Because our work was empirical, and even if it is based on a very large set of real-world data, it will remain provisory, until improved by others. It is important to note two limitations in our study. First, the relationships in Table 5 were developed on catchments with limited interannual memory (in the sense of de Lavenne et al., 2022). Second, aridity was computed using the Oudin et al. (2005) formula for potential evaporation, and the use of other formulas might require a recalibration of the model parameters.

### 6 Acknowledgements

The authors would like to acknowledge the many individuals that worked to make available the hydrological datasets used in this paper. Special thanks are due to Charles Perrin, Matteo Rosales and Guillaume Thirel for their suggestions.

### 7 Funding

This research has been funded in part by the Agence Nationale de la Recherche (projects CIPRHES ANR-20-CE04-0009 and DRHYM ANR-22-CE56-0007).

373

## 8 Author contributions

- VA: conceptualization and writing, GMG: computations, figures, discussion, AL:
- computations, discussion, JL: discussion, writing (review and editing)

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

| 530 | Yang, H., Yang, D., Lei, Z., and Sun, F.: New analytical derivation of the mean annual   |
|-----|------------------------------------------------------------------------------------------|
| 531 | water-energy balance equation, Water Resour. Res., 44, W03410,                           |
| 532 | https://doi.org/03410.01029/02007WR006135, 2008.                                         |
| 533 | Yokoo, Y., Sivapalan, M., and Oki, T.: Investigating the role of climate seasonality and |
| 534 | landscape characteristics on mean annual and monthly water balances, J.                  |
| 535 | Hydrol., 357, 255–269, https://doi.org/10.1016/j.jhydrol.2008.05.010, 2008.              |
| 536 | Zhang, L., and Brutsaert, W: Blending the evaporation precipitation ratio with the       |
| 537 | complementary principle function for the prediction of evaporation. Water Resour.        |
| 538 | Res., 57, e2021WR029729. https://doi.org/10.1029/2021WR029729, 2021.                     |
| 539 | Zhang, Y., Viglione, A., and Blöschl, G.: Temporal scaling of streamflow elasticity to   |
| 540 | precipitation: A global analysis. Water Resour. Res., 58, e2021WR030601.                 |
| 541 | https://doi.org/10.1029/2021WR030601, 2022.                                              |
| 542 |                                                                                          |