# Peer review of "Streamflow elasticity as a function of aridity"

_EGUsphere, 2025_

## Author Comment (AC2)

Dear Dr Anderson,

Thank you very much for reviewing our paper. Please find below (in blue) our answers to the questions you raised.

On behalf of the co-authors,

V. Andréassian

The manuscript titled, "Streamflow elasticity as a function of aridity" empirically validates the relationship between aridity and streamflow elasticity and the proposes a new Budyko type approach to explicitly link elasticity and aridity.

They first compute local elasticities by catchment (multiple regression per basin), then class elasticities for aridity bins (optimising a single triplet of coefficients per aridity class), and finally a regionalized model where coefficients for P and $E_0$ are smooth functions of aridity, constrained by the shape of the Oldekop Budyko derivative. Performance is measured using mean bounded NSE on streamflow anomalies. The key result is that allowing elasticity to vary with aridity via this parametric form improves performance relative to a single global coefficient set, while keeping elasticities in a constrained "physically plausible" range.

The paper was a pleasure to read. The structure, methodological description, and objectives are all very clearly laid out. I believe that the explicit inclusion of aridity in the proposed method is novel and of interest to the community. My comments are all relatively minor and I would recommend this for publication once they are addressed.

**General comments:**

I would describe the quantity calculated here as an absolute marginal sensitivity rather than an elasticity, because it is not a proportional measure.

You are right. We have got used to utilize the word elasticity (sometimes *absolute elasticity*) instead of sensitivity, because the two concepts were so close. Nonetheless we are convinced that utilizing the *absolute elasticity* (i.e. the sensitivity) is preferable, because of the better hydrological interpretation possibilities it offers.

The statement asserted throughout, that bounding elasticity between 0 and [-]1, represents a physically realistic response is, in my opinion, misleading. The sensitivities estimated here are bounded due to the specific structure of the analytical models used, and elasticity very often falls outside of these bounds (in the typical way that its calculated, as well as in the empirical estimates here). The bounds are properties of the specific Budyko formulations and their derivatives, not general physical laws.

On this very point we do not agree: to us the elasticity[1] coefficient represents a "signed yield": $e_{Q/P}$ represents the additional streamflow produced by an additional input of precipitation. Because it is a yield, an additional millimeter of precipitation should (all else being equal) increase streamflow by no more than one millimeter. Similarly, an additional millimeter of potential evapotranspiration should reduce streamflow by no more than one millimeter. Finally, an additional millimeter of synchronous
* * *
[1] we use again elasticity for sensitivity/absolute elasticity to remain coherent with the manuscript

precipitation (P) and potential evapotranspiration ($E_o$), representing the precipitation most easily available for evaporation, should also reduce streamflow by no more than one millimeter. Of course, the sign of the elasticity coefficient is also essential: in the case of $e_{Q/P}$, the coefficient has to be positive (otherwise this would mean that the catchment reacts to an additional input of precipitation by reducing its streamflow, and we consider this unrealistic).

Now are we entitled to claim that an *obvious unrealistic behavior* is *physically unrealistic* ? Is it an *abus de langage[2]*? This is a philosophical question. Let us consider a kind of "proof by contradiction": a catchment where the precipitation elasticity of streamflow would be larger than one would mean a kind of "hydrological chain reaction behavior" that we believe does not exist (at least at the catchment scale and at the annual scale).

Of course, we recognize that when fitting a linear model on annual data, we do observe a few elasticities that are larger than one and have an acceptable *p-value*. But we believe that such things will always happen with large catchment set: the opposite would mean that the dataset has been "cleaned" beforehand.

They assume, implicitly, that additional P cannot reduce ET or storage, that additional PET cannot increase Q, and that storage changes do not play a role. These assumptions may not always hold for annual anomalies (e.g. where storage dynamics or snow processes are important). Values outside of that range in Figure 3 are not necessarily physically unrealistic but may rather represent processes which are not well captured by the assumptions of the model. Discussing this, and explicitly explaining why you assume that 0, 1 are realistic physical boundaries would be helpful.

We agree that exceptions will always remain, where climatic anomalies are amplified. But we should not feel ashamed to accept that our model being extremely simple, it just cannot account for everything and fit every catchment. But even a too-simple model could have physical limits (even if it is too simple to grasp the entire physics)?

The empirical sensitivities presented in the paper are regression coefficients on interannual anomalies. Have you considered under what conditions the Budyko derivative approaches approximate the empirical sensitivities?

Not really, but there is a paper in revision that gives some ideas (Gnann et al).

Elasticity is generally poorly predictable in space (Addor et al., 2018) and while there is a lot of evidence that it relates strongly to the aridity index, the implicit assumption of the regional model is that aridity is the only driver of variation. I still think that what you have done is meritorious, but it would at least be worth discussing that aridity-only regionalisation is a strong simplification.

We will remind it. We like the adjective *meritorious*, we believe that we should all aim at it and not pretend we have produced *great* models.

**Minor comments:**

In general, I would not exclude the points which are not statistically significant from Figure 3. This can be misleading. Please plot them with e.g. high transparency or at least include the full plot in the SI.
* * *
[2] Sorry, we don't know how to translate this French expression

From a sheer statistical point of view, if the coefficient is not significant, it should not be retained. Of course, all limits are arbitrary, but keeping the non-significant 'outliers' on the graph would (i) impose to change the axes limits and pose serious visualisation problems. See a first attempt below: given the number of catchments, we cannot think of a way to plot all points on the same graph.

[Figure]

*Figure 1 : present graph (Figure 3)*

[Figure]

*Figure 2 : alternative graph showing with black crosses the values for which the p-values were larger than the chosen threshold (here 0.05)*

The predictors in use here are, by definition, not independent. Some discussion of collinearity and its impact on coefficient stability would be helpful.

We will add one.

Section 2.3 mentions a "simple grid search algorithm" to calibrate the three coefficients per aridity class, but exact ranges and step sizes aren't given. For such a low-dimensional problem it would be easy to describe the grid explicitly, improving reproducibility.

The grid search in a space of three dimension is quite basic, but we will add a short description.

Please describe the hydrologic memory filter described in the limitations explicitly in the methods section.

We will describe it.

**References**

Addor, N., Nearing, G., Prieto, C., Newman, A. J., Le Vine, N., & Clark, M. P. (2018). A Ranking of Hydrological Signatures Based on Their Predictability in Space. Water Resources Research, 54(11), 8792–8812. https://doi.org/10.1029/2018WR022606

Gnann, S., Anderson, B. J., and Weiler, M.: Uncertainty and non-stationarity of empirical streamflow sensitivities, EGUsphere [preprint], https://doi.org/10.5194/egusphere-2025-4527, 2025.